# Pharmacokinetic Boosting of Kinase Inhibitors

**DOI:** 10.3390/pharmaceutics15041149

**Published:** 2023-04-05

**Authors:** Niels Westra, Daan Touw, Marjolijn Lub-de Hooge, Jos Kosterink, Thijs Oude Munnink

**Affiliations:** 1Department of Clinical Pharmacy and Pharmacology, University Medical Center Groningen, University of Groningen, 9713 GZ Groningen, The Netherlands; 2Pharmaceutical Analysis, Groningen Research Institute of Pharmacy, University of Groningen, 9713 AV Groningen, The Netherlands; 3PharmacoTherapy, Epidemiology & Economics, Groningen Research Institute of Pharmacy, University of Groningen, 9713 AV Groningen, The Netherlands

**Keywords:** pharmacokinetic boosting, pharmacokinetic enhancement, ritonavir, cobicistat, kinase inhibitors, small molecule kinase inhibitors

## Abstract

(1) Introduction: Pharmacokinetic boosting of kinase inhibitors can be a strategy to enhance drug exposure and to reduce dose and associated treatment costs. Most kinase inhibitors are predominantly metabolized by CYP3A4, enabling boosting using CYP3A4 inhibition. Kinase inhibitors with food enhanced absorption can be boosted using food optimized intake schedules. The aim of this narrative review is to provide answers to the following questions: Which different boosting strategies can be useful in boosting kinase inhibitors? Which kinase inhibitors are potential candidates for either CYP3A4 or food boosting? Which clinical studies on CYP3A4 or food boosting have been published or are ongoing? (2) Methods: PubMed was searched for boosting studies of kinase inhibitors. (3) Results/Discussion: This review describes 13 studies on exposure boosting of kinase inhibitors. Boosting strategies included cobicistat, ritonavir, itraconazole, ketoconazole, posaconazole, grapefruit juice and food. Clinical trial design for conducting pharmacokinetic boosting trials and risk management is discussed. (4) Conclusion: Pharmacokinetic boosting of kinase inhibitors is a promising, rapidly evolving and already partly proven strategy to increase drug exposure and to potentially reduce treatment costs. Therapeutic drug monitoring can be of added value in guiding boosted regimens.

## 1. Introduction

Pharmacokinetic boosting, or pharmacokinetic enhancement, is a strategy to optimize the therapeutic properties of a drug [1]. The mechanisms for pharmacokinetic boosting can roughly be divided into four groups [1]: (1) inhibition of hepatic metabolizing enzymes, such as ritonavir inhibiting cytochrome p450 (CYP450); (2) inhibition of drug-specific enzymes, such as carbidopa-inhibiting DOPA decarboxylase; (3) inhibition of bacterial β-lactamase to enhance the antibiotic properties of β-lactam antibiotics, such as clavulanic acid-inhibiting β-lactamase; and (4) absorption enhancement with food.

The first described pharmacokinetic booster was probenecid. During World War II, there was a shortage of penicillin due to the increased demand for penicillin because of the many war casualties. To enable treating more patients with the limited available amount of penicillin, probenecid was used to increase penicillin exposure by decreasing the renal excretion of penicillin [2].

Strategies to enable treating more patients with the same amount of drug are not only of interest in situations of limited availability of drug supplies but can also be of value when drug availability is limited due to high costs. In 1989, the concomitant administration of ketoconazole and cyclosporin was investigated in renal transplant recipients to reduce the high costs associated with cyclosporin use [3]. Cyclosporin boosted with ketoconazole resulted in a reduction of 77% in cyclosporin dosing, while maintaining the immunosuppressive effect of cyclosporin [3].

Pharmacokinetic boosting can be a strategy to enhance the pharmacokinetic profile of a drug. Ritonavir is a protease inhibitor, widely used in the treatment of human immunodeficiency virus (HIV) disease. The potent CYP450-inhibiting properties of ritonavir has led to the use of ritonavir as a pharmacokinetic booster of HIV-protease inhibitors in several HIV combination regimes [4]. In this setting, ritonavir results in a >50 fold increase in saquinavir plasma concentration, and better efficacy of saquinavir is accomplished when given concomitantly [5].

Healthcare costs have been rising worldwide, and the costs are projected to further increase at an annual rate of 3–6% [6]. In particular, newly developed drugs for the treatment of solid and hematologic malignancies are the chief contributors to these rising treatment costs [6,7]. In countries without universal healthcare insurance, the costs for drugs can become unaffordable for individual patients [6]. Pharmacokinetic boosting of expensive drugs can be a promising strategy to reduce rising treatment costs and to allow more patients to benefit from new effective treatments.

Pharmacokinetic boosting, furthermore, is of interest for anticancer drugs with low bioavailability [8,9,10]. In addition to pharmacokinetic boosting, exposure to anticancer drugs with low bioavailability can be increased by various changes to a formulation, such as lipid-based nanocarriers that increase bioavailability [10]. Novel anticancer drugs with better oral bioavailability have been successfully developed, such as cedazuridine/decitabine for the treatment of myeloid malignancies [11].

In the last two decades, many new kinase inhibitors and other oral targeted inhibitor drugs have been approved for treatment of malignant and auto-immune diseases. These often very expensive drugs can be attractive candidates for pharmacokinetic boosting since most are metabolized by CYP3A4 [12], and some have poor absorption that can be enhanced by food.

The aim of this narrative review is to provide answers to the following questions: (1) Which different boosting strategies can be useful in boosting kinase inhibitors? (2) Which kinase inhibitors are potential candidates for either CYP3A4 or food boosting? (3) Which clinical studies on CYP3A4 or food boosting have been published or are ongoing, and what are important lessons from these studies? Additionally, the benefits of boosting, the risks of boosting, clinical trial design and the role of therapeutic drug monitoring (TDM) in pharmacokinetic boosting are discussed.

## 2. Materials and Methods

The selection of potential boosting candidates was based on ATC code, as derived from the WHO ATC/DDD index [13]. Drugs with ATC code L01E, L01XK, L01XG, L01XJ, L01XK, L01XX, L04AA, L01XX52, L01XX59, L01XX62, L01XX73, L01XX77, L04AA29, L04AA32, L04AA37, L04AA44, L04AA46, L04AA49, L04AA56, L04AA59 and D11AH08 were selected to be profiled for pharmacokinetic boosting. This selection contains tyrosine kinase inhibitors, other kinase inhibitors and other oral targeted inhibitor drugs, which will collectively be described as kinase inhibitors for simplicity and recognizability. Parenterally administered drugs and drugs without FDA or EMA approval were excluded, resulting in 85 drugs to be profiled for either CYP3A4 boosting or food boosting. A QuickScan algorithm was used, followed by criteria score-based ranking (Figure 1 and Table 1). Relevant drug characteristics used for selecting candidates and the information in Table 2, Table 3 and Table 4 were retrieved from the Summary of Product Characteristics (SmPC), European Public Assessment Reports (EPAR) and UpToDate [14,15]. The known inhibitory effects of CYP3A4 inhibitors on the potential boosting candidate were retrieved from UpToDate interaction checker by entering the potential boosting candidate in combination with cobicistat [16]. To identify the kinase inhibitors used in the treatment of malignancies and auto-immune disease which are the most eligible candidates for pharmacokinetic boosting, we employed a systematic approach using predefined criteria. For CYP3A4 boosting, the target drug needed to be a substrate for CYP3A4, excluding seven drugs. The remaining 78 drugs were systematically ranked based on criteria which are important for selecting potential boosting candidates. In the case where the boosting candidate forms active metabolites that contribute significantly to the pharmacological effect, the exposure of the active metabolites can be decreased by CYP3A4 boosting, making CYP3A4 boosting less rational. We defined three scores for the active metabolites criterium: 0—no active metabolites or unknown; 1—active metabolites with minor (<10%) contribution to effect; 2—active metabolites with major (>10%) contribution to effect. The second ranking criterium for selecting potential CYP3A4 boosting candidates is the already known effect of strong CYP3A4 inhibitors on the exposure of boosting candidates. This information is derived from drug–drug interaction studies. We defined four scores for the increase in exposure criterium: 0—>200% increase in AUC; 1—100–200% increase in AUC; 2—50–100 % increase in AUC; 5—<50% increase in AUC or unknown. Drugs with mg-based pricing or those for which only one strength of the drug is available are scored 0. Drugs with flat-based pricing for all available strengths are scored 1. As an exception to the aforementioned pricing scores, sonidegib was scored 1 because no dose reduction is possible with the one available strength of sonidegib. The therapeutic value of a drug was not taken into account; it was assumed that all kinase inhibitors have an equivalent therapeutic value.

Kinase inhibitors that are potential candidates for food boosting have been selected using a similar approach. For food boosting, the manufacturer’s label administration recommendation must be to ‘take without food’ or must state that some specific foods cannot be taken in combination with the target drug. Drugs where food has a decreasing effect on the exposure of the target drug are excluded. This QuickScan algorithm for selecting CYP3A4 and/or food boosting candidate is shown in Figure 1.

The PubMed database was used for reviewing available publications. For publications about CYP3A4 boosting, we used the following search query in PubMed: (abemaciclib OR abrocitinib OR acalabrutinib OR adagrasib OR afatinib OR alectinib OR alpelisib OR apremilast OR asciminib OR avacopan OR avapritinib OR axitinib OR baricitinib OR binimetinib OR bosutinib OR brigatinib OR cabozantinib OR capmatinib OR ceritinib OR cobimetinib OR crizotinib OR dabrafenib OR dacomitinib OR dasatinib OR deucravacitinib OR duvelisib OR enasidenib OR encorafenib OR entrectinib OR erdafitinib OR erlotinib OR everolimus OR fedratinib OR futibatinib OR gefitinib OR gilteritinib OR glasdegib OR ibrutinib OR idelalisib OR imatinib OR infigratinib OR ivosidenib OR ixazomib OR lapatinib OR larotrectinib OR lenvatinib OR lorlatinib OR midostaurin OR mobocertinib OR neratinib OR nilotinib OR nintedanib OR niraparib OR olaparib OR osimertinib OR pacritinib OR palbociclib OR pazopanib OR pemigatinib OR pexidartinib OR ponatinib OR pralsetinib OR regorafenib OR ribociclib OR ripretinib OR rucaparib OR ruxolitinib OR selpercatinib OR selumetinib OR sonidegib OR sorafenib OR sotorasib OR sunitinib OR talazoparib OR tepotinib OR tivozanib OR tofacitinib OR trametinib OR tucatinib OR upadacitinib OR vandetanib OR vemurafenib OR venetoclax OR vismodegib OR zanubrutinib) AND (clarithromycin OR cobicistat OR erythromycin OR itraconazole OR ketoconazole OR posaconazole OR ritonavir OR voriconazole OR grapefruit juice).

For publications about food boosting, the following search query was used in PubMed: (Avapritinib OR Cabozantinib OR Erlotinib OR Ibrutinib OR Infigratinib OR Ivosidenib OR Lapatinib OR Nilotinib OR Pazopanib OR Pexidartinib OR Pralsetinib OR Sonidegib OR Sotorasib) AND (food[Title] OR meal[Title] OR low-fat[Title] OR moderate-fat[Title] OR high-fat[Title] OR fasted[Title]).

For ongoing boosting trials, ClinicalTrials.gov was searched with the search terms ‘Oncology’ and ‘CYP3A4′ [17].

## 3. Results

### 3.1. Pharmacokinetic Boosting Strategies Potentially Useful for Kinase Inhibitors

The scope of this review focuses on pharmacokinetic boosting using inhibition of hepatic metabolic enzymes, in particular CYP3A4, and pharmacokinetic enhancement with food.

Metabolism by CYP450 is phase I metabolism that primarily produces hydrophilic structures, which are substrates for phase II metabolism and cleared more easily by the liver, kidney and small intestine [18]. CYP450 is predominantly expressed in the liver but can also be expressed in the kidney, small intestine, lung, brain and can even be found in certain tumor tissues [19,20]. CYP3A4 plays an important role in the bioavailability and exposure of its substrate drugs and is the major CYP enzyme. Approximately 50% of drugs are metabolized by CYP3A4 due to its broad substrate specificity. CYP3A4 activity has inter- and intra-patient variability, which can lead to a variable drug response within and between patients [21]. Inter- and intra-patient variability of CYP3A4 activity can have genetic and epigenetic causes and can also be affected by CYP3A4 induction or inhibition [19,22]. CYP3A5 is another important CYP enzyme, which has overlapping but not identical specificity of substrates with CYP3A4. CYP3A5 is predominantly expressed in the kidneys and lungs and can also be expressed in the liver and intestine. Genetic polymorphisms of CYP3A5 can vary greatly between different ethnic groups, with most patients being CYP3A5 non-expressors [19,23].

Inhibiting CYP3A4 metabolism can potentially lead to an increased exposure of CYP3A4 substrates. To boost the exposure of drugs which are metabolized by CYP3A4, the pharmacokinetic booster drug has to be a strong CYP3A4 inhibitor to effectively boost exposure of the substrate. Examples of strong CYP3A4 inhibitors are clarithromycin, erythromycin, ritonavir, cobicistat, itraconazole, ketoconazole, posaconazole and voriconazole [24].

Several HIV antiretroviral drugs have poor exposure and are metabolized by CYP3A. Atazanavir, darunavir, elvitegravir and lopinavir are HIV antiretroviral drugs boosted by ritonavir or cobicistat to enhance their pharmacokinetic properties. For example, cobicistat enhances the systemic exposure of elvitegravir in the combination elvitegravir/cobicistat/emtricitabine/tenofovirdisoproxil, allowing for a once daily dosing of this single tablet regimen [14]. A more recent example of a similar boosting strategy used in another disease is ritonavir boosting of the SARS-CoV-2 inhibitor nirmatrelvir. When nirmatrelvir is administered alone, it has a T_1/2_ of approximately 2 h, for which it is challenging to maintain the desired plasma concentration of several fold over the in vitro 90% effective concentration. When concomitantly administered with ritonavir, nirmatrelvir has a T_1/2_ of approximately seven hours and an eight-fold increase in exposure, thus enabling a BID dosing regimen [25]. Because boosting strategies can increase the dosing interval and decrease the overall dosage of the substrate drug, boosting can also have an impact on adherence and pill burden [26]. The most widely used agents for CYP3A4 boosting are the pharmacokinetic enhancers ritonavir and cobicistat [4]. Ritonavir and cobicistat are both strong inhibitors of CYP3A4 and can therefore increase the exposure of drugs predominantly metabolized by CYP3A4.

Ritonavir, a HIV protease inhibitor, is an inhibitor of CYP3A4, CYP2D6 and the transporter P-glycoprotein (P-gp), OATP1B1 and an inductor of CYP1A2, CYP2B6, CYP2C8, CYP2C9, CYP2C19 and UGT [14,27]. Ritonavir irreversibly inhibits CYP3A4 [19]. Ritonavir, when used as a pharmacokinetic booster, is dosed 100–400 mg daily. The boosting dose of ritonavir is considered as low-dose ritonavir since therapeutic doses of ritonavir for HIV are with 600 mg BID much higher [26]. Ritonavir has a protein binding of 99%, half-life (T_1/2_) of three to five hours and a distribution volume (Vd) of 20–40 L [14].

Cobicistat was initially developed as an improved version of ritonavir to better facilitate coformulation with other drugs in one tablet. This was possible because cobicistat has a higher water solubility compared to ritonavir [28]. Cobicistat is a structural analogue of ritonavir, but without antiretroviral activity [28]. Cobicistat is a strong CYP3A4 inhibitor and a weak CYP2D6 inhibitor, and furthermore inhibits P-gp, breast cancer resistance protein (BCRP), MATE1, OATP1B1 and OATP1B3. Cobicistat irreversibly inhibits CYP3A4 [29]. The cobicistat label dose is 150 mg once daily. Cobicistat has a protein binding of 98%, T_1/2_ of three to four hours and is known to inhibit renal tubular secretion of creatinine, without affecting the glomerular filtration rate (GFR), which can lead to a slight decrease in the estimated creatinine clearance (CL_cr_) [30]. A comparative overview of ritonavir and cobicistat is presented in Table 2 [14].

Cobicistat and low-dose ritonavir have good pharmacological characteristics for pharmacokinetic boosting of kinase inhibitors. Other strong CYP3A4 inhibitors such as clarithromycin, erythromycin, itraconazole, ketoconazole, posaconazole and voriconazole can also be used as a pharmacokinetic booster, but they have the disadvantage of pharmacological activity and adverse events and can be more expensive. Ketoconazole can result in QT-interval prolongation, and itraconazole is associated with liver toxicity [14]. This makes pharmacologically active CYP3A4 inhibitors less ideal to be used as pharmacokinetic boosters. Cancer patients at risk for invasive fungal disease, however, can benefit from antifungal prophylaxis. In these patients, the antifungal CYP3A4 inhibitor can serve as a two-edged sword combining pharmacokinetic booster and antifungal prophylaxis.

Food is known for its ability to alter the pharmacokinetic properties of a drug [31]. Notable examples are grapefruit juice, Coca-Cola, St. John’s wort and different fasting states. Because grapefruit juice is an inhibitor of CYP3A4, it can potentially boost drugs in a similar manner to cobicistat and ritonavir [32]. Cola is known for its ability to enhance the bioavailability of drugs where low gastric pH is important for the absorption of the drug, especially when the drug is concomitantly given with a proton-pump inhibitor [33]. Different fasting states, such as low-fat meals, moderate-fat meals and high-fat meals, can have effects on the C_max_ and AUC of kinase inhibitors [32]. For the purpose of pharmacokinetic boosting, concomitant intake of certain kinase inhibitors with a high-fat meal can be considered.

### 3.2. Pharmacological Profiling of Candidate Kinase Inhibitors Suitable for Pharmacokinetic CYP3A4 Boosting

The ranking criteria were applied to the selected 78 CYP3A4 boosting candidates and resulted in 10 candidates with score 0 (most attractive candidates); 15 candidates with score 1; 15 candidates with score 2; 7 candidates with score 3; 3 candidates with score 4; 9 candidates with score 5; 11 candidates with score 6; 5 candidates with score 7; and 3 candidates with score 8 (least attractive candidates); see Table 3.

### 3.3. Pharmacological Profiling of Candidate Kinase Inhibitors Suitable for Pharmacokinetic Food Boosting

For food boosting candidates, the manufacturer’s label administration recommendation must be to ‘take without food’ or must state that specific foods that cannot be taken in combination with the target drug. Based on this selection criterium, 65 drugs were excluded. Furthermore, drugs where food has a decreasing effect on the exposure of the target drug were excluded (*n* = 7). The 13 remaining drugs are potential candidates for food boosting (Table 4).

### 3.4. Clinical Evidence and Experience with Pharmacokinetic Boosting of Kinase Inhibitors

#### 3.4.1. Axitinib Boosted with Cobicistat

In a case report, axitinib exposure was boosted by cobicistat [34]. A 54-year-old male, who was diagnosed with metastatic renal cell carcinoma, was previously treated with sunitinib and subsequently everolimus + pazopanib. At disease progression, the patient was switched to axitinib 5 mg BID as a last treatment option. The axitinib dose was escalated after two weeks to 10 mg BID because no serious toxicity was observed. The axitinib C_min_ was measured and was 1.4 microg/L, which is below the reported efficacy threshold of >5 microg/L [35]. Co-medication was screened for drug–drug interactions, and none were found. Screening of CYP450 polymorphisms showed CYP3A4 (*1A/*1B) polymorphism. Axitinib exposure was first boosted by grapefruit juice to inhibit intestinal CYP3A4 metabolism; however, no significant increase in axitinib exposure was seen. Grapefruit juice was subsequently switched to cobicistat 150 mg BID to boost axitinib exposure. After four weeks of axitinib 10 mg BID and cobicistat 150 mg BID, C_2h_ increased by a factor of four, and blood pressure started to slightly increase, but axitinib C_min_ was still below the >5 microg/L threshold. The axitinib and cobicistat dose was further escalated to 10 mg QID and 150 mg QID, respectively. The CT scan showed a decrease in metastasis size, no ascites drainage was needed, and albumin and hemoglobin returned to normal values, which indicates tumor response. After 15 months of ongoing response to treatment with axitinib 10 mg QID boosted by cobicistat 150 mg QID, progressive disease was detected. The patient died two months later as a result of progressive disease. The effect of CYP3A4 (*1A/*1B) polymorphism on the drug exposure of axitinib is unclear. C_min_ of pazopanib and sunitinib, by which the patient was treated initially, were also both below normal values, indicating that the patient had low exposure of pazopanib, sunitinib and axitinib. Because pazopanib, sunitinib and axitinib were all below normal values, the CYP3A4 activity can be a factor explaining the low exposure of these drugs. However, if this is the result of the CYP3A4 (*1A/*1B) polymorphism remains unclear. Based on the observed increase in axitinib concentrations, the authors conclude that boosting axitinib with cobicistat can be a promising and cost-effective strategy for patients with sub-optimal axitinib exposure.

#### 3.4.2. Crizotinib Boosted with Cobicistat

Cobicistat-boosted crizotinib was evaluated in a phase I study in non-small cell lung cancer patients with low crizotinib exposure [36]. Crizotinib has a high interindividual variability of C_min,ss_, which is associated with concentration-dependent variability in overall response rate (ORR). In the quartile with the lowest C_min,ss_, the ORR is 24–47%; in the quartile with the highest C_min,ss_, the ORR is 60–75%. Patients in the lower quartile versus the remaining three quartiles are associated with a higher hazard ratio of 3.2 [37]. The study hypothesis was that patients in the lower quartile for C_min,ss_ could have a better outcome when crizotinib exposure is boosted with cobicistat. Patients who received a minimum of 14 days of standard care with crizotinib and had a C_min,ss_ ≤ 310 ng/mL were eligible. Only one patient was included because the study was prematurely terminated for ethical reasons after approval of more potent second-generation drugs. Before cobicistat 150 mg QD was added to the patient using crizotinib, blood samples for pharmacokinetic analysis were drawn. C_min,ss_ was measured seven days after concurrent treatment of cobicistat with crizotinib. After 14 days of concurrent treatment of cobicistat with crizotinib, blood samples for pharmacokinetic analysis were drawn. The AUC_0–12h_ and C_min,ss_ increased by 78% and 164%, respectively, when crizotinib was boosted by cobicistat. No serious adverse events (AEs) were observed. The authors conclude that cobicistat can be a promising and non-expensive strategy to increase crizotinib exposure and suggest that this strategy might also be of value for other kinase inhibitors.

#### 3.4.3. Erlotinib Boosted with Ritonavir

A phase I open-label crossover study was performed to evaluate the feasibility of erlotinib exposure boosting by ritonavir and thereby reducing the erlotinib dose to achieve cost savings [38]. Non-small cell lung cancer patients who received erlotinib 150 mg QD for a minimum of eight days were eligible for inclusion. Nine patients were included for the primary analysis of the study. After a minimum of eight days of treatment with erlotinib 150 mg QD, blood samples for pharmacokinetic analysis were drawn. The erlotinib dose was subsequently reduced to 75 mg QD, and patients were treated for seven days with the reduced dose. After these seven days, ritonavir 200 mg QD was added to erlotinib 75 mg QD, and patients were treated for seven days. Subsequently, blood samples were drawn for pharmacokinetic analysis, and patients were switched back to standard of care erlotinib 150 mg QD. The geometric mean ratio (GMR) of erlotinib 150 mg QD compared to erlotinib 75 mg + ritonavir 200 mg QD for AUC_0–24h_, C_max_ and C_min_ were 0.99 (CI 95% 0.58–1.69, *p* = 0.545), 0.91 (CI 95% 0.55–1.49, *p* = 0.500) and 1.06 (CI 95% 0.59–1.93, *p* = 0.150), respectively. A statistically significant decrease in active metabolites OSI-413 and OSI-420 was observed. No grade ≥ 3 AEs were reported. The coefficient of variability (CV%) was 58–162% for erlotinib (+metabolites) and was 86–443% for ritonavir-boosted erlotinib (+metabolites). The authors expected that the CV% for erlotinib + ritonavir would be lower compared to erlotinib alone. The higher CV% for erlotinib + ritonavir could be the result of a shift in erlotinib metabolism. When erlotinib + ritonavir are concurrently administered, the main metabolism route shifts from CYP3A4 to CYP1A2 (and other isoforms), because ritonavir strongly inhibits the major CYP3A4 metabolism route. CYP1A2 expression is known to differ between patients and thus can be a factor explaining the increased CV%. Firm conclusions about the increased CV% are, however, difficult to be drawn because of the limited number of patients included in the study. The authors conclude that pharmacokinetic exposure of erlotinib 150 mg QD compared to erlotinib 75 mg QD + ritonavir 200 mg QD is equivalent, and erlotinib boosting can be a strategy to reduce an erlotinib dose by 50% and thus save treatment costs.

#### 3.4.4. Ibrutinib Boosted with Itraconazole

In a randomized placebo-controlled crossover study with healthy volunteers, the boosting effect of itraconazole on ibrutinib exposure was evaluated [39]. The aim was to reduce the high interindividual variability of ibrutinib and to reduce treatment costs. Participants (*n* = 11) were randomly assigned to either the cohort ibrutinib 15 mg + itraconazole or ibrutinib 140 mg + placebo. Subjects were given itraconazole 200 mg BID or placebo BID on day 1, and on days 2–4 itraconazole 200 mg QD or placebo QD. On day 3, subjects who received placebo were given 140 mg ibrutinib, and subjects who received itraconazole received ibrutinib 15 mg. After a washout period of four weeks, subjects were enrolled in the crossover cohort. Ibrutinib 15 mg + itraconazole had a similar exposure when compared to ibrutinib 140 mg + placebo; the GMR of AUC_0-∞_ and C_max_ was 1.07 (CI 90% 0.77–1.49; *p* = 0.719) and 0.94 (CI 90% 0.68–1.30, *p* = 0.727), respectively. The geometric CVs of AUC_0-∞_ and C_max_ for ibrutinib 15 mg boosted with itraconazole were 0.55 and 0.53, respectively, and for ibrutinib 140 mg + placebo 1.04 and 0.99, respectively, indicating reduced interindividual variation for boosted ibrutinib. According to the manufacturer’s dose recommendation when ibrutinib is concomitantly administered with a strong CYP3A4 inhibitor, the advice is to adjust the dose from 420 mg or 560 mg to 140 mg. However, the results in this study suggest a dose reduction of 90%. The authors conclude that ibrutinib boosting with itraconazole reduces interindividual variability and increases ibrutinib exposure and this enables improved dosing accuracy while achieving 90% cost savings. Annual cost savings with boosted ibrutinib in the United States are projected to be more than $10,000 per patient.

#### 3.4.5. Imatinib Boosted with Grapefruit Juice

In an open-label, non-randomized, within-group crossover study, imatinib was boosted with grapefruit juice to ascertain if dose reduction of imatinib is feasible to reduce treatment costs [40]. Four patients with chronic myeloid leukemia (CML) who were treated with imatinib 400 mg QD for more than six months were eligible for inclusion. Blood samples for pharmacokinetic analysis were drawn when patients were using imatinib 400 mg QD. After two to three months, 250 mL Tropicana^®^ grapefruit juice QD was added to imatinib 400 mg QD and administered for seven consecutive days. After these seven days of concurrent treatment with grapefruit juice and imatinib, blood samples for pharmacokinetic analysis were drawn. The median C_min_ was 1080 ng/mL (range: 1060–1360 ng/mL) and 1102 ng/mL (range: 772–1450 ng/mL) for imatinib 400 mg and imatinib 400 mg in combination with grapefruit juice, respectively. The median C_max_ was 2495 ng/mL (range: 2380–2680 ng/mL) and 2455 ng/mL (range: 1870–2750 ng/mL) for imatinib 400 mg and imatinib 400 mg in combination with grapefruit juice, respectively. No serious AEs were observed. Pharmacokinetics of imatinib 400 mg QD compared to imatinib 400 mg QD boosted with grapefruit juice were not significantly different. A possible explanation is that this is due to the fact grapefruit juice predominantly inhibits intestinal CYP3A4 and to a lesser extent hepatic CYP3A4. Imatinib bioavailability is almost 100%, thus only inhibiting the intestinal CYP3A4 has little effect. The study was prematurely terminated because no significant effect of grapefruit juice on imatinib exposure was observed.

#### 3.4.6. Lapatinib Boosted with Ketoconazole

The phase I dose escalation study of lapatinib evaluated different dose-escalating strategies for lapatinib in patients with HER2 positive breast cancer to enhance the exposure of lapatinib [41]. Patients with HER2 overexpression advanced breast cancer and cardiac ejection fraction ≥ 50% were eligible for inclusion. The study included a total of 41 patients divided into 10 cohorts. The cohorts one to six had a predefined dose-escalating strategy without any pharmacokinetic boosting agent. After an interim analysis, it was decided that, in cohorts seven to ten, the lapatinib exposure was boosted by a pharmacokinetic enhancer. Concomitant intake with food and with or without the CYP3A4 inhibitor ketoconazole were chosen as exposure enhancement strategies. Cohorts eight to ten were concurrently treated with lapatinib BID or QID, with food and with ketoconazole 200 mg BID. A total of 12 patients were in cohorts eight to ten. Lapatinib plasma concentration blood samples were drawn at baseline and four hours after the morning dose. Food did not increase the exposure of lapatinib. Concomitant administration of lapatinib with ketoconazole increased lapatinib exposure 2.7 fold.

#### 3.4.7. Nilotinib Boosted with Food

The NiFo study evaluated if the nilotinib dose could be reduced when concurrently administered with food to reduce the complexity of the dosing regimen and to achieve cost savings [42]. CML patients in the chronic phase (*n* = 15) who had had at least three months of treatment with nilotinib prior to the study were included. The first four days of the study, patients received standard of care nilotinib 300 mg BID in fasted state, followed by seven days of nilotinib 200 mg BID concurrently administered with food. Morning meals were low-fat, evening meals were medium-fat, and on days 8 and 11, the evening dose was taken with high-fat meals. Blood samples for pharmacokinetic analysis were drawn on days 1, 3, 8 and 11. The GMR of the morning dose of AUC_0–12h_, C_max_ and C_min_ was 0.89 (CI 90% 0.81–0.98), 0.90 (CI 90% 0.8–1.02) and 0.88 (CI 90% 0.84–0.92), respectively, and were within acceptance limits for bioequivalence. The GMR of the evening dose of AUC_0–12h_, C_max_ and C_min_ was 0.84 (CI 90% 0.73–0.97), 0.8 (CI 90% 0.68–0.93) and 1.06 (CI 90% 0.92–1.22), respectively. The GMR of C_min_ was within acceptance limits for bioequivalence, the GMR of AUC_0–12h_ and C_max_ were not. Nilotinib 200 mg BID with food was well tolerated, and patient-reported symptom burden was lower compared to nilotinib 300 mg BID standard of care. Bioequivalence for C_min_ was reached; AUC_0–12h_ and C_max_ were not bioequivalent. Nilotinib efficacy is, however, associated with C_min_; patients with a C_min_ above the threshold of ≥619 ng/mL have a higher major molecular response at three months [43]. Boosting nilotinib 200 mg BID with food can therefore still be a viable option, especially when guided with therapeutic drug monitoring (TDM).

#### 3.4.8. Osimertinib Boosted with Cobicistat

The effect of cobicistat on osimertinib exposure was investigated in the proof-of-concept OSIBOOST study [44]. The aim of this study was to evaluate if osimertinib exposure could be increased by cobicistat and if the boosting effect was stable over time. Cobicistat was selected as a CYP3A4 inhibitor because of its strong inhibition of CYP3A4, lack of off-target effects and its wide use in clinical practice as a boosting agent in antiretroviral therapies. A total of 11 non-small cell lung cancer patients that had a low osimertinib exposure of C_min,ss_ ≤ 195 ng/mL were included. Patients were initially treated with osimertinib standard of care (10 patients at 80 mg QD and 1 patient at 160 mg QD), and blood samples were drawn for pharmacokinetic analysis. After the first blood sampling day, patients were given cobicistat 150 mg QD in combination with osimertinib. The second pharmacokinetic sampling day was scheduled 22–26 days after the start of the concurrent usage of osimertinib with cobicistat 150 mg QD. After the second pharmacokinetic sampling day, patients could opt to stop the treatment of cobicistat, to continue with the concurrent use if adequate boosting was reached or to continue with the concurrent use where cobicistat was stepwise escalated to 150 mg BID or QID for patients with osimertinib C_min,ss_ ≤ 195 ng/mL. The primary outcome was the change in AUC_0–24,ss_ of osimertinib and its metabolite AZ5104 when boosted with cobicistat compared to osimertinib alone. Secondary outcomes included CYP3A4 and CYP3A5 polymorphisms, AEs and osimertinib C_min,ss_ as a surrogate marker of AUC_0–24,ss_ for patients who continued in the study after the first phase. During concurrent use of cobicistat, the AUC_0–24,ss_ of all patients increased with a mean AUC_0–24,ss_ increase of 60%. The mean AUC_0–24,ss_ increase had a relatively broad range of 19–192%. The mean AUC_0–24,ss_ increase in women and men was 73% and 38%, respectively. Three patients had osimertinib C_min_ ≤ 195 ng/mL with concurrent use of cobicistat; their cobicistat dosage was consequently escalated to 150 mg BID. In one patient, osimertinib AUC_0–24,ss_ decreased after cobicistat escalation, while in the other two patients the osimertinib AUC_0–24,ss_ increased. In the one patient where the osimertinib AUC_0–24,ss_ decreased with concurrent use of cobicistat 150 mg BID, the cobicistat dosage was further escalated to 150 mg QID. This decreased the osimertinib AUC_0–24,ss_ even further to an overall increase of 1% compared to baseline. No serious AEs were observed. Concurrent use of cobicistat and osimertinib increased the exposure of osimertinib and its metabolite AZ5104 in all patients and can be an option to reduce the osimertinib dose. The added value of measuring the osimertinib metabolite AZ5104 was limited. The interindividual variation of the boosting effect of cobicistat on osimertinib exposure is challenging when composing a one-fits-all concept. Although interindividual variation of the boosting effect was relatively high, the boosting effect was constant within patients, paving the way to an integrated TDM-guided approach to cobicistat-boosted osimertinib.

#### 3.4.9. Pazopanib Boosted with Continental Breakfast

The DIET study evaluated whether it was feasible to enhance the exposure of pazopanib with food to reduce the pazopanib dose in patients with renal cell carcinoma [45]. The study consisted of two parts. The first part was a pharmacokinetic dose-finding study to confirm bioequivalence of 800 mg QD in fasted state compared to pazopanib 600 mg QD taken with a continental breakfast. Nineteen patients were enrolled for the first part of the study. Patients received pazopanib 800 mg for 14 days in fasted state, followed by pazopanib 600 mg QD with a continental breakfast. GMR of steady state AUC_0–24h_, C_max_ and C_min_ was 1.09 (CI 90% 1.02–1.17), 1.12 (CI 90% 1.04–1.20) and 1.10 (CI 90% 1.02–1.18), respectively. The second part of the study was conducted to evaluate gastrointestinal toxicity and patient preference for pazopanib 600 mg QD combined with continental breakfast compared to pazopanib 800 mg QD in a fasted state. Patients (*n* = 78) were initially enrolled in the second part of the study and randomly assigned to either pazopanib 800 mg QD in fasted state or pazopanib 600 mg QD with a continental breakfast. After four weeks, patients were switched to the opposite regimen. Pazopanib 600 mg QD with continental breakfast was preferred by 68% of the patients. Pazopanib 800 mg QD compared to pazopanib 600 mg QD + continental breakfast was bioequivalent; gastrointestinal AEs were comparable in both groups. Pazopanib + continental breakfast can achieve a total cost savings of approximately $8500 per patient for metastatic renal cell carcinoma and approximately $3800 per patient for soft tissue sarcoma in the Netherlands.

#### 3.4.10. Tofacitinib Boosted with Cobicistat

The PRACTICAL study was performed to evaluate the feasibility of boosting tofacitinib exposure by cobicistat, reducing 50% of the dose and saving 50% in treatment costs [46]. The study was an open-label, non-randomized, within group crossover study, where bioequivalence of tofacitinib 5 mg QD boosted with cobicistat 150 mg QD was compared to the standard of care tofacitinib 5 mg BID. Patients with rheumatoid arthritis or psoriatic arthritis who received a minimum of 14 days of standard care with tofacitinib were eligible for inclusion. A total of 25 patients were included for the primary analysis of the study. After ≥14 days of tofacitinib treatment, blood samples for pharmacokinetic analysis were drawn from patients receiving tofacitinib 5 mg BID standard of care. Patients were subsequently switched to tofacitinib 5 mg QD + cobicistat 150 mg QD. Between two to six weeks after the switch to cobicistat-boosted tofacitinib, blood samples for pharmacokinetic analysis were drawn. Medication adherence was monitored in a medication diary. Patient preference was evaluated after the second pharmacokinetic sampling day. GMR of tofacitinib C_avg,ss_ for tofacitinib 5 mg BID compared to tofacitinib 5 mg QD + cobicistat 150 mg QD was 0.85 (CI 90%: 0.75–0.96) and was therefore not pharmacokinetically bioequivalent according to the EMA acceptance interval. Interindividual variability expressed as relative bioavailability was 21% (residual standard error 73%) for the boosted regimen versus 32.2% (residual standard error 30.9%) for the non-boosted regimen. Disease activity remained stable, and no serious AEs were observed. The once-daily tofacitinib 5 mg QD + cobicistat 150 mg QD regimen, compared to tofacitinib 5 mg BID, was preferred by 56% of the patients. The tofacitinib 5 mg QD + cobicistat regimen can potentially achieve annual cost savings of approximately €6500 per patient in the European Union and approximately €21,500 in the United States until the patent expiry date of 2028.

#### 3.4.11. Venetoclax and Ibrutinib Boosted with Itraconazole

In a case report, venetoclax and ibrutinib were boosted with itraconazole to save treatment costs [47]. A 22-year-old man with acute myeloid leukemia (AML) was treated with a 75% reduced dose of venetoclax 100 mg QD with itraconazole 100 mg BID as the boosting drug. No complications developed, and the patient achieved complete response, incomplete hematological recovery and a nondetectable minimal residual disease. The patient subsequently received an allogeneic stem cell transplantation. At 40 days after the stem cell transplantation, the patient developed a grade III steroid-refractory acute graft versus host disease (GvHD) which was eventually treated with ibrutinib. The ibrutinib dose was 75% reduced compared to the normal dose and was 140 mg QD boosted with itraconazole 100 mg BID. After three weeks, the patient achieved a complete response of GvHD. After 11 months, the patient remained completely responsive, and ibrutinib was tapered. CYP3A4 boosting of venetoclax and ibrutinib to reduce treatment costs by 75% was concluded to be a promising strategy, and subsequent prospective clinical trials were initiated. A total of approximately $10,900 in cost savings was achieved in this patient by boosting venetoclax and ibrutinib.

#### 3.4.12. Venetoclax Boosted with Posaconazole

The venetoclax–posaconazole drug–drug interaction study evaluated which dose adjustment is necessary when venetoclax is concurrently administered with posaconazole [48]. Patients (*n* = 12) diagnosed with AML and eligible for inclusion were included. On days one to five, patients received a venetoclax ramp-up from 20–200 mg and intravenous decitabine 20 mg/m^2^. On days 6 to 20, patients received venetoclax 400 QD. On days 21 to 28, patients received a reduced venetoclax dose of either 50 mg or 100 mg in combination with posaconazole 300 mg BID on day 21, and days 22 to 28 posaconazole 300 mg QD. Six patients received the venetoclax 100 mg QD dose reduction, and five patients received the venetoclax 50 mg QD dose. The duration of the posaconazole treatment was determined to be 8 days to reach steady state. Blood samples for pharmacokinetic analysis were drawn on day 20 and day 28. In patients who received venetoclax 50 mg QD in combination with posaconazole, the mean AUC_0–24h_ and C_max_ increased by 76% and 53%, respectively, when compared to venetoclax 400 mg QD alone. In patients who received venetoclax 100 mg QD in combination with posaconazole, the mean AUC_0–24h_ and C_max_ increased by 155% and 93%, respectively, when compared to venetoclax 400 mg QD alone. Coadministration of posaconazole in combination with either venetoclax 50 mg QD or 100 mg QD was overall well tolerated. The venetoclax dose should be reduced by at least 75% when co-administered with posaconazole.

#### 3.4.13. Venetoclax Boosted with Grapefruit Juice

Venetoclax was boosted with grapefruit juice in a patient with AML who could not afford the regular dose of 400 mg QD [49]. Treatment started with venetoclax 100 mg QD in combination with 200 mL grapefruit juice TID. Venetoclax C_max_ was measured weekly to ascertain adequate exposure to venetoclax and to reduce toxicity. The venetoclax C_max_ was 1440 ng/mL and 1920 ng/mL on day 7 and day 14 after receiving the combination venetoclax 100 mg QD and grapefruit juice 200 mL TID, respectively. The venetoclax C_max_ was inside the efficacy boundary, as stated by the authors, of 1000–3000 ng/mL. The patient was in remission for at least five cycles of 28 days, and no serious AEs were observed. Boosting venetoclax with grapefruit juice to make the treatment more affordable was concluded to be safe and effective for this patient. The venetoclax-associated monthly costs were reduced from 38,880 RMD yuan (approx. €5281) to 9720 RMD Yuan (approx. €1319).

The aforementioned studies are summarized in Table 5, to provide a comparative overview.

### 3.5. Clinical Trials Currently in Progress

Five ongoing trials were identified from ClinicalTrials.gov [17] where pharmacokinetic boosting had to be intentional and could not be a regular drug–drug interaction trial. Pharmacokinetic boosting in these ongoing trials is done to either save treatment costs or to investigate potential therapeutic benefits when the target drug is boosted. The studies evaluate efficacy of the boosted regimen with different outcomes, only the PROACTIVE study additionally include pharmacokinetic parameters as an outcome. Table 6 presents an overview of the currently ongoing boosting trials.

## 4. Discussion

Pharmacokinetic boosting of kinase inhibitors is a rapidly evolving field, as indicated by the increasing evidence from published clinical studies and ongoing trials. Pharmacokinetic boosting can be a promising strategy for increasing exposure of anticancer drugs, which was also indicated by two previous review articles [8,9]. In the sections below, the most important aspects of the clinical boosting trials and pharmacokinetic boosting in general are discussed.

### 4.1. Benefits of Pharmacokinetic Boosting of Kinase Inhibitors

Increasing the bioavailability of a drug can theoretically lead to a decrease in inter-patient variability [50]. Furthermore, genetic polymorphisms of metabolizing enzymes such as CYP450 can account for variable exposure to the drug, resulting in variability in therapeutic responses between patients [44]. By inhibiting the CYP450 enzyme responsible for the metabolism of the target drug, the inter-patient variability of plasma concentrations can theoretically decrease [26]. In the study where ibrutinib is boosted with itraconazole, the interindividual variability in exposure was decreased by pharmacokinetic boosting [39].

Boosting of expensive drugs has the potential to drastically reduce treatment costs. The high cost-saving potential of the pharmacokinetic boosting of kinase inhibitors is quantified or projected in some trials [39,45,46,47,49]. For some boosted kinase inhibitors, the clinical evidence for pharmacokinetic boosting safety and efficacy is already substantial. The highly reduced venetoclax dose in combination with CYP3A4 inhibitors has proven to be safe without compromising its efficacy [48], making venetoclax boosting an attractive strategy for treating patients where financial resources are limited [51].

For some kinase inhibitors, pharmacokinetic boosting can, in addition to cost savings, also result in an optimized dosing regimen. The study with cobicistat-boosted tofacitinib indicated that with pharmacokinetic boosting the standard BID tofacitinib regimen might be reduced to a once-daily regimen when tofacitinib is combined with cobicistat [46].

### 4.2. Risks and Disadvantages of Pharmacokinetic Boosting of Kinase Inhibitors

When drugs are intentionally off-label-boosted, this can incorporate additional risks and disadvantages compared to the standard non-boosted regimen. The exposure of the target drug can increase or decrease compared to the normal dosing regimen and can therefore increase or decrease the efficacy and toxicity of the target drug. Ascertaining bioequivalence based on the EMA bioequivalence guideline of the boosted versus non-boosted regimen is therefore important [52].

When a strong CYP3A4 inhibitor is concurrently administered with the target drug, the CYP3A4 inhibitor can also interact with the comedication of a patient. CYP3A4 inhibition of interacting comedication can lead to increased toxicity or decreased efficacy of these interacting drugs. To mitigate this risk, it is advisable to screen the comedication for drug–drug interactions before starting with the boosted regimen. When a drug–drug interaction is found, ideally an alternative for the interacting drug that is not affected by the boosting drug should be considered. When no adequate alternative is available or appropriate, a dose adjustment can be considered. When the risk of drug–drug interactions is inappropriate and no alternative or dose adjustment can be found, the patient cannot participate in a boosting regimen. When a CYP3A4 inducer is present in a patient’s comedication, this can interfere with the pharmacokinetic boosting agent. Ideally the CYP3A4 inducer is switched to another drug; otherwise, the patient is unlikely to be suitable for pharmacokinetic boosting.

Inter-patient variability can decrease in a boosted regimen when compared to the non-boosted regimen. However, two clinical studies found an increased variability for boosted kinase inhibitors [38,44]. This increased variability can be due to the fact the metabolism shifts from predominantly CYP3A4-mediated metabolism to another CYP450 enzyme responsible for the metabolism with high variable activity. Unexplored causes for this increased variability have to be investigated. However, the increased variability is not necessarily a problem since some drugs do not have a strong exposure response or exposure toxicity relation. In contrast, for drugs with a known small therapeutic range, increased variability can be more problematic. To decrease the possible increased inter- and intra-individual pharmacokinetic variability, an individual TDM approach is a possibility to mitigate this risk.

Furthermore, drug-specific and disease-related risks can also be important factors. Aside from the aforementioned risks, important exclusion criteria for risk mitigation can be impairment of the gastrointestinal tract that can alter absorption, renal impairment, hepatic impairment, pregnancy and lactation and severe therapy-associated toxicity.

### 4.3. Factors for Selecting a Pharmacokinetic Boosting Candidate

Aside from our ranked boosting candidates as shown in Table 3, additional factors for selecting a boosting candidate have to be taken into account. Some kinase inhibitors are also substrates for transporters such as P-gp and BCRP. P-gp is expressed in multiple organs such as the small intestine, liver, kidney and the blood-brain barrier [53]. When drugs which are substrate for P-gp are boosted with a drug that is also a P-gp inhibitor or inducer, this can cause suboptimal exposure or increased toxicity of the target drug. It is therefore important that drug transporters are also taken into account when selecting a boosting candidate. Some kinase inhibitors are inhibitors of their own metabolism (auto-inhibition) which can complicate a boosting strategy [54]. However, boosting kinase inhibitors with auto-inhibiting properties compel the need to guide therapy with TDM.

When the primary goal of pharmacokinetic boosting is to reduce treatment costs, the pricing of the target drug is an important factor that has to be taken into account. Drugs can be priced based on formulation strength or can be flat-priced with the same price for different doses [55]. Expensive drugs that are priced based on formulation strength (linear pricing) are more suitable for pharmacokinetic boosting for economic purposes than flat-priced drugs [55]. However, drug manufacturers can change the pricing structure to flat-based pricing to maximize revenues as a reaction to lower dosing regimens. This was, for example, implemented by the manufacturer of ibrutinib in the United States after a study showed equivalent efficacy of a lower ibrutinib dosing regimen [56]; however, after public objection, this decision was reversed [57]. A changed pricing structure by a manufacturer might therefore be a risk for a boosting strategy. Manipulation of dosage forms could counter the issues presented by flat-based pricing, as was performed by altering the sorafenib formulation [58]. Furthermore, it is important to consider and estimate the projected therapeutic value for upcoming years and to indicate possible shifts in overlapping indications. The study with crizotinib, for example, was prematurely terminated because the first-choice treatment option shifted from crizotinib to alectinib, making crizotinib boosting less clinically relevant [36]. In addition, factors such as total costs per patient, annual volume and the patent expiration date of the candidate drug are relevant factors when pharmacokinetic boosting is performed for economic purposes.

### 4.4. Clinical Trial Design of Studies Validating Pharmacokinetic Boosting

Pharmacokinetic boosting of kinase inhibitors might be applied in individual cases with low exposure at high doses and suboptimal disease control. However, for pharmacokinetic boosting of a certain drug to become more widely applied or standard-of-care, the result of pharmacokinetic boosting first has to be validated in a clinical trial. The first step in a pharmacokinetic boosting study is to determine the bioequivalent dose of the boosted regimen versus the non-boosted regimen. The kinase inhibitor dose with a booster can be estimated using pharmacokinetic or drug–drug interaction data. The best starting point is the manufacturer’s recommendation for dose adjustment when co-administered with interacting drugs, such as a strong CYP3A4 inhibitor. Drug–drug interaction studies are generally available for new drugs with potential drug–drug interactions based on preclinical pharmacology. Non-linearity in drug–drug interaction studies can be an important factor for determining a good dose adjustment, such as a four-fold increase in drug exposure when concomitantly administered with a strong CYP3A4 inhibitor that does not necessarily translate to a dose reduction of 75%.

The estimated kinase inhibitor dose in combination with a booster drug has to be compared to the standard dose without booster drug for at least pharmacokinetic and safety endpoints. A bioequivalence clinical trial can be a method to compare the boosted and non-boosted regimens. The EMA bioequivalence guideline advises designing the clinical trial as a randomized, two-period, two-sequence, single-dose crossover with a wash-out period between the two periods of a minimum of five half-lives [52]. For the purpose of pharmacokinetic boosting, the EMA-recommended trial design, however, can be amended to better accommodate the specific needs of a boosting trial. Blinding the study drugs incorporates logistical issues and can complicate the clinical trial, increasing the costs of the clinical trial. Pharmaceutical companies are not keen to sponsor boosting trials because boosting potentially reduces the revenue of highly profitable drugs. Because funding of a fully blind, randomized, placebo-controlled clinical trial can be challenging, an open-label design is probably the most suitable for a clinical boosting study. For the purpose of ascertaining bioequivalence of boosted versus non-boosted, multiple doses can be given to actual patients versus the single dose in healthy volunteers proposed by the EMA guideline [52]. The number of participants in the clinical trial should be based on the sample size calculation, with a minimum of 12. To determine bioequivalence at steady-state, the AUC_(0-t),ss_, C_max,ss_ and T_max,ss_ have to be measured. C_min_ can also be a useful parameter to determine because it can be a surrogate marker for exposure. The acceptance level of the 90% confidence interval of the ratio of boosted versus non-boosted AUC_(0-t)_ and C_max,ss_ is ≥80% and ≤125%. This 80–125% acceptance interval can be tightened for drugs with a narrow therapeutic index and can be widened for drugs with high (>30%) intra-subject variability. For kinase inhibitors with no exposure–toxicity relation, it might be considered to only use the lower boundary of the acceptance interval to ensure that the boosted exposure is at least the exposure of the non-boosted dose. The T_max_ only needs to be statistically tested when a rapid onset of the tested drug is of clinical importance. When the first few participants have completed the bioequivalence trial, an interim analysis to review the preliminary bioequivalence results can be useful. When in the preliminary data of the first few participants no bioequivalence is observed and the effect of boosting is higher or lower than expected, the remaining participants can be switched to a higher or lower dose for the remainder of the bioequivalence study part. When bioequivalence is consequently ascertained for the boosted versus non-boosted regimen, the efficacy of the boosted regimen can then be compared against the standard of care.

When pharmacokinetic equivalence of the boosting regimen has been established, the second part of a boosting trial can consist of comparing the boosted regimen with the non-boosted standard of care for safety and efficacy. The level of evidence needed depends on the effect of the booster on the exposure [59]. When the boosted regimen has a bioequivalent exposure compared to the non-boosted regimen, a study on efficacy does not necessarily have to be performed [59]. However, alternative regimens where bioequivalence is already determined are rarely used in clinical practice [59]. Even if the boosted regimen is bioequivalent, further study of efficacy can be considered to strengthen the evidence so that the boosted regimen has a greater chance of being implemented in clinical practice. When efficacy evaluation is considered in a pharmacokinetic boosting trial, it has the potential to have a cost-neutral clinical trial budget when the projected high savings of the drug are realized during the trial. Patients can be randomly assigned to the boosted regimen (intervention) arm or the non-boosted standard-of-care (control) arm. Another option is to compare the boosted regimen (intervention) arm to a real-life cohort of the same population as a control arm. Outcomes that can be considered include overall survival, AUC_(0-t)_, C_max,ss_, and T_max,ss_ for bioequivalence and possibly other pharmacokinetic parameters, efficacy endpoints based on the disease for which the drug is used, and in general quality of life and event-free survival, safety endpoints such as AEs and early mortality and exploratory endpoints such as inter- and intra-individual variability, patient preference, cost-savings and medication adherence.

### 4.5. Role of Therapeutic Drug Monitoring in Pharmacokinetic Boosting of Kinase Inhibitors

Therapeutic drug monitoring (TDM) is usually practiced to optimize the dose of a drug or minimize toxicity of a drug based on measured serum of plasma concentration and using estimated or calculated individual pharmacokinetic parameters. Kinase inhibitors are mainly dosed with a fixed dose derived from the maximum tolerated dose from phase I and II clinical studies [60], which focus more on toxicity and less on efficacy. Therefore, a clear exposure–response relationship cannot always be established; however, this does not necessarily mean there is no exposure-response relationship present. Drugs without exposure–response and exposure–toxicity relationships are unlikely to benefit from a TDM-guided approach in routine clinical practice because it is not clear to which extent the dose has to be adjusted.

Drugs with an exposure–response or exposure–toxicity relationship are good candidates to incorporate TDM in the boosting regimen, especially for drugs where TDM is already proven to be of benefit [61,62]. Dose adjustments for boosting can be estimated based on the known pharmacokinetic outcomes associated with efficacy and toxicity [49]. TDM guidance can therefore help individualizing the appropriate dose and potentially reduce exposure variability. TDM guidance in this population can also lead to less toxicity because overexposure of the drug of interest is detected early, and the dose can be decreased accordingly. Drugs without a strong exposure–efficacy relationship, but with a known small exposure–toxicity relationship, can also be guided by TDM. When a patient, for example, presents with unexplainable toxicity, a drug concentration can be measured to exclude drug overexposure as a possible explanation for the observed toxicity.

In cases of boosting, bioavailability is increased and/or systemic clearance is decreased. TDM can be used to individually titrate the dose and to ascertain bioequivalence of a boosted regimen versus a non-boosted regimen. Further, it can be used to monitor medication adherence. The added value of TDM of kinase inhibitors ideally has to be established in a clinical trial, which is challenging to perform and can be expensive [63].

A drawback to implement TDM as part of the care is the required infrastructure, such as a validated analytical method for measuring drug concentrations, which can be lacking. Furthermore, the required infrastructure can be expensive to develop and maintain. It might be helpful when there is an overview of laboratories which can perform TDM on kinase inhibitors. TDM of kinase inhibitors can also introduce regulatory issues because dose adjustments based on TDM can be off-label. Still, guiding boosted kinase inhibitors with TDM can be of added value [62], but these disadvantages have to be recognized.

## 5. Conclusions

Pharmacokinetic boosting of kinase inhibitors is a promising, rapidly evolving and already partly proven strategy, which has the potential to reduce interindividual variability, reduce pill burden and drastically reduce treatment costs of expensive kinase inhibitors. Current evidence consists of prospective clinical trials and some case reports, and several clinical trials are ongoing. Ascertaining bioequivalence should be enough evidence to implement a boosting regimen in clinical practice. TDM in routine clinical practice can be of added value in guiding boosted regimens.

## Figures and Tables

**Figure 1 pharmaceutics-15-01149-f001:**
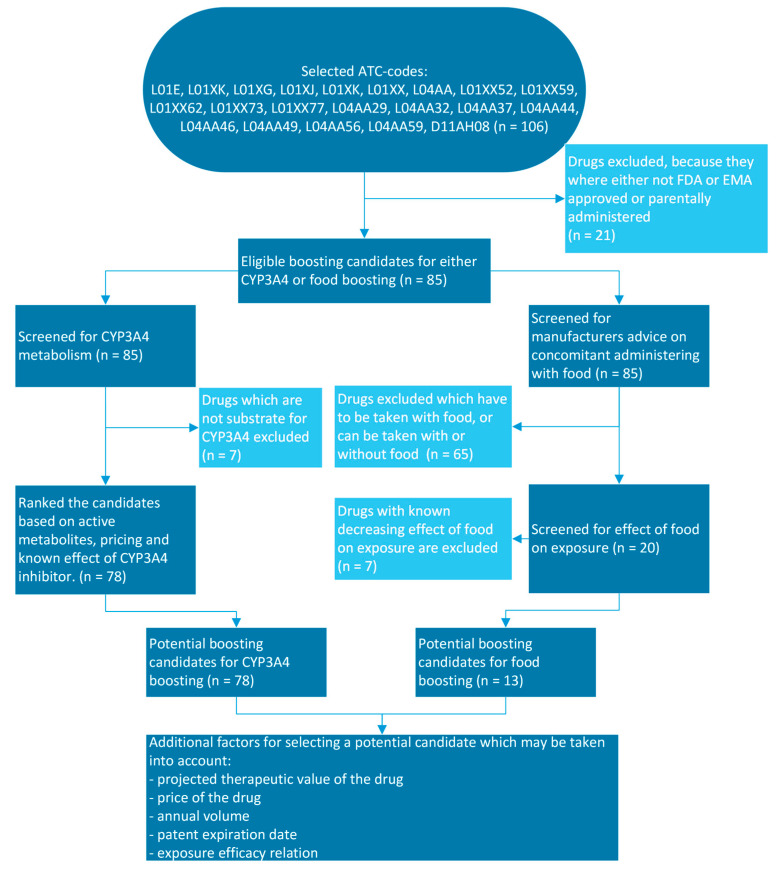
QuickScan algorithm for selecting either CYP3A4 or food boosting candidates.

**Table 1 pharmaceutics-15-01149-t001:** Scoring criteria of potential CYP3A4 boosting candidates.

Active Metabolites	Known Effect of CYP3A4 Inhibitor on AUC	Pricing of Different Drug Strengths
0—no active metabolites or unknown	0—>200% increase in AUC	0—mg-based pricing or one strength available
1—minor active metabolites (<10% responsible for efficacy)	1—100–200% increase in AUC	1—flat-based pricing for all available strengths
2—major active metabolites (>10% responsible for efficacy)	2—50–100% increase in AUC	
	5—<50% or unknown increase in AUC	

**Table 2 pharmaceutics-15-01149-t002:** Comparison of relevant drug properties of ritonavir and cobicistat. Relevant drug characteristics were retrieved from the Summary of Product Characteristics (SmPC), European Public Assessment Reports (EPAR) and UpToDate [14,15].

	Ritonavir	Cobicistat
Antiviral activity	Yes, HIV protease inhibitor; inhibits HIV-1 and HIV-2	No
Dosage as pharmacokinetic enhancer	100 mg or 200 mg; QD or BID	150 mg QD
Protein binding	99%	98%
Half-life	3–5 h	3–4 h
Distribution volume	20–40 L	Not known
Metabolized by	CYP3A4 and to lesser extent CYP2D6	CYP3A4 and to lesser extent CYP2D6
Inhibitor of	CYP3A4 (strong), CYP2D6 (minor), P-gp and OATP1B1	CYP3A4 (strong) CYP2D6 (minor), P-gp, BCRP, MATE1, OATP1B1, OATP1B3
Inducer of	CYP1A2, CYP2B6, CYP2C8, CYP2C9, CYP2C19, UGT	-
In vitro CYP3A4 inhibition duration	Irreversible	Irreversible

**Table 3 pharmaceutics-15-01149-t003:** Selection of potential CYP3A4 boosting candidates (*n* = 78). Drugs are ranked according to ranking score (low → high) and on alphabet within the same ranking score. Candidates with the lowest ranking score have the most potential for CYP3A4 boosting. Relevant drug characteristics were retrieved from the Summary of Product Characteristics (SmPC), European Public Assessment Reports (EPAR) and UpToDate [14,15].

Drug	Pricing of Different Strengths	Metabolism	Substrate of Transporters	Bioavailability	CYP3A4 Inhibitory Drug	Effect on AUC (Fold Change)	Effect on C_max_ (Fold Change)	Ranking Score
Adagrasib	One strength available	CYP3A4, however at steady state, adagrasib inhibits its own CYP3A4 metabolism, which allows CYP2C8, CYP1A2, CYP2B6, CYP2C9, and CYP2D6 to contribute to metabolism.	BCRP/ABCG2	-	Itraconazole	4	2.4	0
Bosutinib	Strength-based pricing	CYP3A4 to primarily inactive metabolites M2, M5 and M6	-	34% when administered with food	Ketoconazole	8.6	5.2	0
Cobimetinib	One strength available	CYP3A4	P-gp/ABCB1	46%	Itraconazole	6.7	3.2	0
Duvelisib	Strength-based pricing	CYP3A4	BCRP/ABCG2	42%	Ketoconazole	4	1.7	0
Encorafenib	One strength available	CYP3A4 and to a lesser extent by CYP2C19 and CYP2D6	P-gp/ABCB1	≥86% of the dose is absorbed	Posaconazole	3	1.7	0
Fedratinib	One strength available	CYP3A4, CYP2C19, and flavin-containing monooxygenase 3 (FMO3)	OATP1B1/1B3 (SLCO1B1/1B3)	-	Ketoconazole	3.1–3.9	1.9	0
Lapatinib	One strength available	CYP3A4 and CYP3A5 and to a lesser extent by CYP2C19 and CYP2C8 to metabolites	BCRP/ABCG2 and P-gp/ABCB1	-	Ketoconazole	3.6	2.1	0
Larotrectinib	Strength-based pricing	CYP3A4; forms a metabolite (activity unknown)	BCRP/ABCG2 and P-gp/ABCB1	34%	Itraconazole	4.3	2.8	0
Pralsetinib	One strength available	CYP3A4 and to a lesser extent CYP2D6 and CYP1A2	BCRP/ABCG2 and P-gp/ABCB1	-	Itraconazole	3.5	1.8	0
Zanubrutinib	One strength available	CYP3A4	-	-	Ketoconazole	3.8	2.6	0
Avacopan	One strength available	CYP3A4	-	-	Itraconazole	2.19	1.87	1
Avapritinib	Flat pricing	CYP3A4 and CYP3A5 and to lesser extent CYP2C9, which forms the metabolite M690	-	-	Itraconazole	7	-	1
Axitinib	Strength-based pricing	CYP3A4/5 and to a lesser extent CYP1A2, CYP2C19 and UGT1A1		58%	Ketoconazole	2.1	1.5	1
Brigatinib	Strength-based pricing	CYP2C8 and CYP3A4	BCRP/ABCG2 and P-gp/ABCB1	-	Itraconazole	2.01	-	1
Ceritinib	One strength available	CYP3A4	P-gp/ABCB1	-	Ketoconazole	2.9	1.2	1
Crizotinib	Flat pricing	CYP3A4/5	P-gp/ABCB1	43%	Ketoconazole	3.2	1.7	1
Dasatinib	Strength-based pricing	CYP3A4, flavin-containing mono-oxygenase-3 (FOM-3) and UGT to an active metabolite (minor role in the efficacy)	-	-	Ketoconazole	4.8	3.6	1
Glasdegib	Strength-based pricing	CYP3A4 and to a lesser extent CYP2C8 and UGT1A9	BCRP/ABCG2	77%	Ketoconazole	2.4	1.4	1
Ibrutinib	Strength-based pricing	CYP3A and to a lesser extent CYP2D6 to form active metabolite PCI-45227(minor role in the efficacy)	-	2.9%	Ketoconazole	24	29	1
Ivosidenib	One strength available	CYP3A4 and to a lesser extent the N-dealkylation and hydrolytic pathways	P-gp/ABCB1	-	Itraconazole	2.7	No change	1
Nilotinib	Flat pricing	CYP3A4 to primarily inactive metabolites	P-gp/ABCB1	50%	Ketoconazole	3	1.8	1
Olaparib	One strength available	CYP3A4	P-gp/ABCB1	-	Itraconazole	2.7	1.4	1
Ribociclib	One strength available	CYP3A4 to metabolites M13, M4 and M1 (minor role in the efficacy)	-	-	Ritonavir	3.2	1.7	1
Selpercatinib	Strength-based pricing	CYP3A4	BCRP/ABCG2	73%	Itraconazole	2.3	1.3	1
Venetoclax	Strength-based pricing	CYP3A to form the major metabolite M27 (BCL-2 inhibitory activity 58-fold lower)	BCRP/ABCG2 and P-gp/ABCB1	-	Ritonavir	6.1–8.1	2.3–2.4	1
Acalabrutinib	One strength available	CYP3A4 enzymes and to a lesser extent glutathione conjugation and amide hydrolysis to form active metabolite ACP-5862	BCRP/ABCG2 and P-gp/ABCB1	25%	Itraconazole	5.1	3.7–3.9	2
Dabrafenib	Strength-based pricing	CYP2C8 and CYP3A4 to form active metabolite hydroxy-dabrafenib	BCRP/ABCG2 and P-gp/ABCB1	95%	Ketoconazole	1.71	-	2
Erlotinib	Strength-based pricing	CYP3A4 and to a lesser extent CYP1A1, CYP1A2, and CYP1C to form metabolites (activity unknown)	-	60% without food, 100% with food	Ketoconazole	1.69	1.52	2
Everolimus	Flat pricing	CYP3A4 and forms six metabolites with minor activity	P-gp/ABCB1	30%	Ketoconazole	15	3.9	2
Gefitinib	One strength available	CYP3A4 and to a lesser extent CYP2D6. Forms metabolites	BCRP/ABCG2	60%	Itraconazole	1.16–1.78	1.32–1.51	2
Gilteritinib	One strength available	CYP3A4 to form active metabolites M17, M16 and M10 (minor role in the efficacy)	P-gp/ABCB1	-	Itraconazole	2.2	1.2	2
Infigratinib	Strength-based pricing	CYP3A4 and to a lesser extent FMO3 to form active metabolites BHS697 and CQM157	BCRP/ABCG2 and P-gp/ABCB1	-	Itraconazole	7.22	2.64	2
Mobocertinib	One strength available	CYP3A to form active metabolites AP32960 and AP32914	P-gp/ABCB1	37%	Itraconazole	6.3	2.9	2
Neratinib	One strength available	CYP3A4 and flavin-containing monooxygenase to form active metabolites M3, M6, M7, and M11	P-gp/ABCB1	-	Ketoconazole	4.8	3.2	2
Pacritinib	One strength available	CYP3A4 and forms the 2 major metabolites M1 and M2 (activity unknown)	-	-	Clarithromycin	1.8	1.3	2
Pazopanib	One strength available	CYP3A4 and P-gp/ABCB1 and to a lesser extent by CYP1A2, CYP2C8 and BCRP/ABCG2	BCRP/ABCG2 and P-gp/ABCB1	-	Ketoconazole	1.66	1.45	2
Pexidartinib	One strength available	CYP3A4 and glucuronidation via UGT1A4 to form an inactive metabolite	-	-	Itraconazole	1.73	1.48	2
Sonidegib	One strength available	CYP3A4	-	<10%	Ketoconazole	2.2	1.5	2
Tofacitinib	Flat pricing	CYP3A4 and CYP2C19 to form inactive metabolites	-	74%	Itraconazole	2.04	1.15	2
Upadacitinib	Strength-based pricing	CYP3A4	-	-	Ketoconazole	1.75	1.7	2
Entrectinib	Flat pricing	CYP3A4 to form the active metabolite M5	-	-	Itraconazole	6	1.7	3
Idelalisib	Flat pricing	Aldehyde oxidase and CYP3A, which forms major metabolite GS-563117, to a lesser extent UGT1A4	BCRP/ABCG2 and P-gp/ABCB1	-	Ketoconazole	1.8	No change	3
Nintedanib	Flat pricing	Hydrolytic cleavage by esterases to inactive metabolite BIBF 1202, which is further UGT 1A1, UGT 1A7, UGT 1A8, and UGT 1A10 to BIBF 1202 glucuronide, and to a lesser extent CYP3A4	OCT1 and P-gp/ABCB1	5%	Ketoconazole	1.6	1.8	3
Palbociclib	Flat pricing	CYP3A4 and SULT2A1	-	46%	Itraconazole	1.87	1.34	3
Pemigatinib	Flat pricing	CYP3A4	BCRP/ABCG2 and P-gp/ABCB1	-	Itraconazole	1.88	1.17	3
Ponatinib	Flat pricing	CYP3A4 and to a lesser extent CYP2C8, CYP2D6, and CYP3A5	BCRP/ABCG2 and P-gp/ABCB1	-	Ketoconazole	1.78	1.47	3
Ripretinib	One strength available	CYP3A4 and to a lesser extent CYP2C8 and CYP2D6 to form active metabolite DP-5439(activity unknown)	BCRP/ABCG2 and P-gp/ABCB1	-	Itraconazole	1.99	1.36	3
Abemaciclib	Flat pricing	CYP3A4 to form active metabolites M2, M20, M18 and M1	BCRP/ABCG2 and P-gp/ABCB1	45%	Clarithromycin	2.5	-	4
Midostaurin	One strength available	CYP3A4 to form active metabolites CGP62221 and CGP52421	-	-	Ketoconazole	10.4	1.8	4
Sunitinib	Strength-based pricing	CYP3A4 to form active metabolite SU12662	-	-	Ketoconazole	1.51	1.49	4
Apremilast	One strength available	CYP3A4 and to a lesser extent CYP1A2 and CYP2A6	P-gp/ABCB1	73%	-	-	-	5
Baricitinib	Strength-based pricing	CYP3A4	BCRP/ABCG2 and P-gp/ABCB1	80%	-	-	-	5
Erdafitinib	Strength-based pricing	CYP2C9 and CYP3A4	P-gp/ABCB1		Itraconazole	1.34	No change	5
Lorlatinib	Strength-based pricing	CYP3A4 and UGT1A4 and to a lesser extent CYP2C8, CYP2C19, CYP3A5 and UGT1A3	-	81%	Itraconazole	1.42	1.24	5
Selumetinib	Strength-based pricing	CYP3A4 and to a lesser extent BCRP/ABCG2, CYP1A2, CYP2A6, CYP2C19, CYP2C9, CYP2E1, CYP3A4, P-glycoprotein/ABCB1, UGT1A1 and UGT1A3	BCRP/ABCG2 and P-gp/ABCB1	62%	Itraconazole	1.49	1.19	5
Sotorasib	One strength available	CYP3A4, CYP3A5 and CYP2C8	-	-	-	-	-	5
Tucatinib	Strength-based pricing	CYP2C8 and to a lesser extent CYP3A	BCRP/ABCG2	-	-	-	-	5
Vemurafenib	One strength available	BCRP/ABCG2 and CYP3A4 and to a lesser extent P-gp/ABCB1	BCRP/ABCG2 and P-gp/ABCB1	64%	Itraconazole	1.4	1.4	5
Ruxolitinib	Flat pricing	CYP3A4 and to lesser extent CYP2C9 to form active metabolites	-	-	Ketoconazole	1.91	1.33	5
Alpelisib	Flat pricing	Chemical and enzymatic hydrolysis to form its metabolite and to a lesser extent CYP3A4	BCRP/ABCG2	-	-	-	-	6
Asciminib	Flat pricing	CYP3A4, UGT2B7 and UGT2B17	BCRP/ABCG2 and P-gp/ABCB1	-	Clarithromycin	1.36	1.19	6
Cabozantinib	Flat pricing	CYP3A4	-	-	Ketoconazole	1.36	-	6
Capmatinib	Flat pricing	CYP3A4 and aldehyde oxidase	P-gp/ABCB1	>70%	Itraconazole	1.42	No change	6
Enasidenib	Flat pricing	CYP1A2, CYP2B6, CYP2C8, CYP2C9, CYP2C19, CYP2D6, CYP3A4, UGT1A1, UGT1A3, UGT1A4, UGT1A9, UGT2B7 and UGT2B15	-	57%	-	-	-	6
Futibatinib	unknown	CYP3A, and to a lesser extent CYP2C9 and CYP2D6	BCRP/ABCG2 and P-gp/ABCB1	-	Itraconazole	1.41	1.51	6
Ixazomib	Flat pricing	CYP3A4, CYP1A2, CYP2B6, CYP2C8, CYP2D6, CYP2C19 and CYP2C9	P-gp/ABCB1	58%	-	-	-	6
Lenvatinib	Flat pricing	CYP3A4 and aldehyde oxidase	BCRP/ABCG2 and P-gp/ABCB1	-	-	-	-	6
Rucaparib	Flat pricing	CYP2D6 and to a lesser extent CYP1A2 and CYP3A4	BCRP/ABCG2 and P-gp/ABCB1	36%	-	-	-	6
Tepotinib	One strength available	CYP3A4 and CYP2C8 to form an active metabolite	P-gp/ABCB1	71.6%	-	-	-	6
Tivozanib	Flat pricing	CYP3A4	-	-	Ketoconazole	1.12	-	6
Alectinib	One strength available	CYP3A4 to form active metabolite M4	-	37%	-	-	-	7
Imatinib	Strength-based pricing	CYP3A4 and to a lesser extent CYP1A2, CYP2D6, CYP2C9 and CYP2C19 to form active metabolite CGP74588	OCT1 and P-gp/ABCB1	98%	Ketoconazole	1.4	1.26	7
Regorafenib	One strength available	CYP3A4 and UGT1A9 to form active metabolites M2 and M5	-	-	Ketoconazole	1.33	-	7
Sorafenib	One strength available	CYP3A4 and UGT1A9 to form an active metabolite	-	38–49%	Itraconazole	No change	No change	7
Vandetanib	Strength-based pricing	CYP3A4 to form active metabolites N-desmethyl vandetanib and vandetanib-N-oxide	-	-	Itraconazole	1.09	No change	7
Abrocitinib	Flat pricing	CYP2C19 and to a lesser extent CYP2C9, CYP3A4 and CYP2B6 to form active metabolites 3-hydroxypropyl abrocitinib and 2-hydroxypropyl abrocitinib	OAT1/3	60%	-	-	-	8
Dacomitinib	Flat pricing	Oxidation and glutathione conjugation and by CYP2D6 and CYP3A4 to form active metabolite O-desmethyl dacomitinib	BCRP/ABCG2	80%	-	-	-	8
Osimertinib	Flat pricing	CYP3A4 to form active metabolites Z7550 and AZ5104	BCRP/ABCG2 and P-gp/ABCB1		Itraconazole	1.24	.8	8

**Table 4 pharmaceutics-15-01149-t004:** Final selection of potential food boosting candidates (*n* = 13). Relevant drug characteristics were retrieved from the Summary of Product Characteristics (SmPC), European Public Assessment Reports (EPAR) and UpToDate [14,15].

Drug	Pricing of Different Strengths	Bioavailability	Food Effect
Avapritinib	Flat pricing	-	AUC and C_max_ increased 1.29 and 1.59-fold, respectively, when administered with a high-fat, high-calorie meal
Cabozantinib	Flat pricing	-	AUC and C_max_ increased 1.57 and 1.41-fold, respectively, when administered with a high-fat meal
Erlotinib	Strength-based pricing	60% without food	Absorption 60% in fasted state, food increases absorption to 100%
Ibrutinib	Strength-based pricing	2.9%	AUC and C_max_ increased two-fold and two- to four-fold, respectively, when administered with a high-fat, high-calorie meal
Infigratinib	Strength-based pricing	-	AUC and C_max_ increased 1.8- to 2.2-fold and 1.6- to 1.8-fold, respectively, when administered with a high-fat, high-calorie meal
Ivosidenib	One strength available	-	AUC and C_max_ increased 1.24 and 1.98-fold, respectively, when administered with a high-fat meal
Lapatinib	One strength available	-	AUC increased three- to four-fold when administered with food
Nilotinib	Flat pricing	50%	Bioavailability increased 1.82-fold when administered 30 min after a high-fat meal
Pazopanib	One strength available	-	AUC increased two-fold when administered with a high-fat or low-fat meal
Pexidartinib	One strength available	-	AUC and C_max_ increased two-fold when administered with a high-fat meal
Pralsetinib	One strength available	-	AUC and C_max_ increased 2.22 and 2.04-fold, respectively, when administered with a high-fat meal
Sonidegib	One strength available	<10%	AUC increased seven- to eight-fold when administered with a high-fat meal
Sotorasib	One strength available	-	AUC increased 1.25-fold when administered with a high-fat meal

**Table 5 pharmaceutics-15-01149-t005:** Overview of kinase inhibitor boosting studies.

Target Drug	Boosting Agent	Study Aim	Study Design	Outcomes	Results	Conclusion	Reference
Axitinib	Cobicistat 150 mg QID	Boost axitinib exposure with cobicistat	Case report, one patient	C_min_	Axitinib 10 mg QID + cobicistat 150 mg QID resulted in a 15-month stable response	Boosting axitinib with cobicistat can be a promising strategy to boost patients with sub-optimal axitinib exposure.	[34]
Crizotinib	Cobicistat 150 mg QD	Patients with low crizotinib exposure (C_min,ss_ ≤ 310 ng/mL) were boosted with cobicistat	Open-label, non-randomized, within group crossover study, one patient	Change in AUC_0–24,ss_ and Cmin_0–24,ss_	The AUC and C_min,ss_ increased by 78% and 164% respectively when crizotinib was boosted by cobicistat.	Cobicistat enhanced the exposure of crizotinib. Only one patient was enrolled because the next-generation ALK inhibitor alectinib was approved for the treatment of the same population with better outcomes.	[36]
Erlotinib	Ritonavir 200 mg QD	Bioequivalence of erlotinib 150 mg QD compared to erlotinib 75 mg QD + ritonavir 200 mg QD to save treatment costs	Open-label, non-randomized, within group crossover study, nine patients	GMR of AUC_0–24h_, C_max_ and C_min_	GMR of erlotinib 150 mg QD vs. erlotinib 75 mg + ritonavir 200 mg QD for AUC_0–24h_, C_max_ and C_min_ were 0.99 (CI 95% 0.58–1.69, *p* = 0.545), 0.91 (CI 95% 0.55–1.49, *p*= 0.500) and 1.06 (CI 95% 0.59–1.93, *p* = 0.150), respectively.	Erlotinib 150 mg QD compared to erlotinib 75 mg + ritonavir 200 mg is bioequivalent and can be a strategy to reduce the erlotinib dosage by 50% and thus save treatment costs.	[38]
Ibrutinib	Itraconazole 200 mg BID	Evaluate exposure of Ibrutinib 15 mg + itraconazole compared to ibrutinib 140 mg + placebo	Randomized placebo-controlled crossover study with 11 healthy volunteers	GMR of AUC_0-∞_ and C_max_	GMR of ibrutinib 15 mg + itraconazole vs. ibrutinib 140 mg + placebo AUC_0-∞_ and C_max_ were 1.07 (CI 90% 0.77–1.49; *p* = 0.719) and 0.94 (CI 90% 0.68–1.30, *p* = 0.727), respectively, the GMR CVs of AUC_0-∞_ and C_max_ for ibrutinib 15 mg boosted + itraconazole were 0.55 and 0.53, respectively, and for ibrutinib 140 mg + placebo 1.04 and 0.99, respectively.	The interindividual variability of exposure of ibrutinib is high; boosting with itraconazole and a reduced dose of ibrutinib could lower the interindividual variability. Boosting with itraconazole is cost-effective and can potentially reduce the treatment costs associate with ibrutinib by 90%. Cost savings in the United States are projected to be more than $10,000 annually per patient.	[39]
Imatinib	Grapefruit juice	Evaluate whether reduction of imatinib is feasible to reduce treatment costs with grapefruit juice	Open-label, non-randomized, within group crossover study, four patients	C_min_ and C_max_	The median C_min_ was 1080 ng/mL (range: 1060–1360 ng/mL) and 1102 ng/mL (range: 772–1450 ng/mL) for imatinib 400 mg and imatinib 400 mg in combination with grapefruit juice, respectively. The median C_max_ was 2495 ng/mL (range: 2380–2680 ng/mL) and 2455 ng/mL (range: 1870–2750 ng/mL) for imatinib 400mg and imatinib 400mg in combination with grapefruit, juice respectively.	Pharmacokinetic of imatinib 400 mg QD compared to imatinib 400 mg QD boosted with grapefruit juice was not significantly different. The study was prematurely terminated because no significant effect of grapefruit juice on imatinib pharmacokinetics was observed.	[40]
Lapatinib	Ketoconazole 200 mg BID	Evaluate dose-escalating strategies for lapatinib	Phase I dose escalation study, 12 patients in the cohorts boosted with ketoconazole	Lapatinib concentration	Concomitant administration of lapatinib + ketoconazole increased lapatinib exposure 2.7 fold.	Lapatinib exposure can be enhanced by ketoconazole.	[41]
Nilotinib	Food; low-fat, medium-fat and high-fat meals	Evaluate whether nilotinib exposure can be enhanced with food	Open-label, non-randomized, within-group crossover study, 15 patients	GMR of AUC_0–12h_, C_max_ and C_min_	The GMR of the morning dose of AUC_0–12h_, C_max_ and C_min_ was 0.89 (CI 90% 0.81–0.98), 0.90 (CI 90% 0.8–1.02) and 0.88 (CI 90% 0.84–0.92), respectively, and were within acceptance limits for bioequivalence. The GMR of the evening dose of AUC_0–12h_, C_max_ and C_min_ was 0.84 (CI 90% 0.73–0.97), 0.8 (CI 90% 0.68–0.93) and 1.06 (CI 90% 0.92–1.22), respectively.	Bioequivalence for C_min_ was reached; AUC0–12h and C_max_ were not bioequivalent. Nilotinib efficacy is associated with C_min_, meaning that nilotinib 200 mg BID with food can still be a viable option.	[42]
Osimertinib	Cobicistat 150 mg QD	Patients with low osimertinib exposure (C_min,ss_ ≤ 195 ng/mL) were boosted with cobicistat	Open-label, non-randomized, within-group crossover study, 11 patients	Change in AUC_0–24,ss_ (primary) and C_min_	The mean AUC_0–24,ss_ increase with cobicistat was 60%	Concurrent use of cobicistat and osimertinib increased the exposure of osimertinib and its metabolite AZ5104 in all patients and can be an option to reduce the osimertinib dose.	[44]
Pazopanib	Food; continental breakfast	Evaluate whether pazopanib exposure can be enhanced with food	Open-label, randomized, within-group crossover study, 19 patients in part 1, 78 patients in part 2	GMR of AUC_0–24h_, C_max_, C_min_, gastrointestinal toxicities and patient preference	The GMR of steady state AUC_0–24h_, C_max_ and C_min_ was 1.09 (CI 90% 1.02–1.17), 1.12 (CI 90% 1.04–1.20) and 1.10 (CI 90% 1.02–1.18), respectively.	Pazopanib 800 mg QD compared to pazopanib 600 mg QD + continental breakfast is bioequivalent, gastrointestinal AEs were comparable in both groups. Pazopanib + continental breakfast can achieve savings of approximately $8500 per patient for metastatic renal cell carcinoma and approximately $3800 per patient for soft tissue sarcoma in the Netherlands.	[45]
Tofacitinib	Cobicistat 150 mg QD	Bioequivalence of tofacitinib 5 mg BID compared to tofacitinib 5 mg QD + cobicistat 150 mg QD	Open-label, non-randomized, within-group crossover study, 25 patients	GMR of C_avg,ss_	GMR of tofacitinib C_avg,ss_ for tofacitinib 5 mg BID vs. tofacitinib 5 mg QD + cobicistat 150 mg QD was 85% (CI 75–96%)	Tofacitinib 5 mg BID compared to tofacitinib 5 mg QD + cobicistat 150 mg are not pharmacokinetically bioequivalent. Disease activity remained stable, indicating similar efficacy. The tofacitinib 5 mg QD + cobicistat can potentially achieve annual cost savings of approximately €6500 per patient in the European Union and approximately €21,500 in the United States until the patent expiry date of 2028.	[46]
Venetoclax and ibrutinib	Itraconazole 100 mg BID	Evaluate whether a 75% dose reduction of venetoclax and ibrutinib is feasible when co-administered with itraconazole	Case report, one patient	Efficacy	A 22-year-old man was successfully treated with a 75% reduced dose of venetoclax 100 mg QD + itraconazole 100 mg BID and ibrutinib 75% reduced dose of 140 mg QD + itraconazole 100 mg BID.	CYP3A4 boosting with itraconazole to reduce the treatment costs of venetoclax and ibrutinib can be a promising strategy. A total of approximately $10,900 in cost savings was achieved in this patient by boosting venetoclax and ibrutinib. More research to validate this hypothesis is warranted; especially prospective studies are required.	[47]
Venetoclax	Posaconazole 300 mg QD	Evaluate which dose adjustment is necessary when venetoclax is concurrently administered with posaconazole	Drug–drug interaction study, 12 patients	AUC_0–24h_ and C_max_	Venetoclax 50 mg QD + posaconazole increased the mean AUC_0–24h_ and C_max_ by 76% and 53%, respectively, vs. venetoclax 400 mg QD alone. Venetoclax 100 mg QD + posaconazole increased the mean AUC_0–24h_ and C_max_ by 155% and 93% vs. venetoclax 400 mg QD alone.	The venetoclax dose should be reduced by at least 75% when co-administered with posaconazole.	[48]
Venetoclax	Grapefruit juice	Evaluate whether venetoclax 100 mg QD could be boosted by grapefruit juice so that the therapy becomes more affordable	Case-report, one patient	C_max_ and efficacy	The venetoclax C_max_ was 1440 ng/mL and 1920 ng/mL on day 7 and day 14 after receiving the combination venetoclax 100 mg QD and grapefruit juice 200 mL TID. The venetoclax C_max_ was inside the efficacy boundary of 1000 ng–3000 ng/mL. The patient was in remission for at least five cycles of 28 days; no serious AEs were observed.	Boosting venetoclax to make the treatment more affordable with grapefruit juice was safe and effective for this patient. The venetoclax-associated monthly costs were reduced from 38,880 RMD yuan (approx. €5281) to 9720 RMD Yuan (approx. €1319).	[49]

**Table 6 pharmaceutics-15-01149-t006:** Overview of currently ongoing trials on kinase inhibitor boosting (*n* = 5).

Study Name	Target Drug	Boosting Agent	Study Aim	Study Design	Outcomes	NCT Number
Cytochrome P450 Inhibition to Decrease Dosage of Dasatinib for Chronic Myelogenous Leukemia	Dasatinib	Ketoconazole	Investigate whether a 75% dasatinib dose reduction when boosted with ketoconazole is feasible to reduce treatment costs	Phase II open-label single-arm study with 15 participants	Primary: Cytogenetic and molecular response rates and AEs	NCT05638763
Efficacy and Safety of Low-dose Ibrutinib and Itraconazole in Chronic Graft Versus Host Disease	Ibrutinib	Itraconazole	Investigate whether a 75% ibrutinib dose reduction when boosted with Itraconazole is feasible to reduce treatment costs	Phase II open-label single-arm study with 13 participants	Primary: Overall response rate and AEs	NCT05348096
Pharmacokinetic Boosting of Olaparib to Improve Exposure, Tolerance and Cost-effectiveness (PROACTIVE)	Olaparib	Cobicistat	Ascertain bioequivalence of olaparib 300 mg BID vs. olaparib 100 mg BID + cobicistat 150 mg BID to reduce treatment costs	Part 1: bioequivalence in a cross-over olaparib vs. boosted olaparib. Part 2: non-inferiority of olaparib vs. boosted Olaparib, 160 participants	Primary: AUC_0–12h_, progression-free survival, number of dose reductions as a measure of toxicity	NCT05078671
A Study of Extending Relugolix Dosing Intervals Through Addition of Itraconazole or Ritonavir in Prostate Cancer Patients	Relugolix	Itraconazole or ritonavir	Investigate safety and efficacy of relugolix when combined with itraconazole or ritonavir to extend dosing interval of relugolix to reduce treatment costs	Phase Ib in 100 participants	Primary: testosterone suppression	NCT05679388
Low-dose Venetoclax and Azacitidine as Front-line Therapy in Newly Diagnosed AML	Venetoclax	Itraconazole	Investigate whether a 75% venetoclax dose reduction when boosted with Itraconazole is feasible to reduce treatment costs	Phase II open-label single-arm study with 15 participants	Primary: number of patients who are hospitalized, number of deceased patients in predefined time frames	NCT05048615

## Data Availability

Not applicable.

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
