# Peer review of "Pharmacokinetic Boosting of Kinase Inhibitors"

_pharmaceutics, 2023, doi:10.3390/pharmaceutics15041149_

Round 1

Reviewer 1 Report

The manuscript ‘Pharmacokinetic boosting of kinase inhibitors’ by Westra et al. has been reviewed.

CYP3A4 metabolism constitutes a mechanism that significantly reduces the biodistribution of drugs, including kinase inhibitors.

The review describes the problem of phase I biotransformations, and focuses on 13 clinical studies aimed at increasing drug availability through CYP3A4 inhibition.

The argument is of interest, the authors list many interesting studies, but in my opinion a final perspective will be beneficial to attract more interest from the readers.

The manuscript presents some imprecisions/typos, please find below just few examples:

-Typos: line 42 is, line 210 booster, line 258 dose, line 456 of, line 469 to….

- The quality of the figure 1 should be improved.

- References should be added in the tables, as done in Table 5.

In my opinion, the reported work should be accepted after minor revisions noted.

Reviewer 2 Report

In this narrative review, Westra et al. describe the pharmacokinetic boosting of kinase inhibitors. This review effectively answers several important questions: 1) What is PK boosting and why might kinase inhibitors benefit? 2) Which kinase inhibitors are most likely to benefit? 3) What has previously been done to examine the feasibility of boosting kinase inhibitors? 4) What should be done in the future? This paper includes narrative descriptions of several recent trials that were attempting to achieve the goal of boosting a kinase inhibitor.

Overall, this paper is very good and of interest to the field. I especially appreciated the rigorous description of their search strategy, ranking algorithm, and strategy to establish a boosting regimen as clinically equivalent. I think a few minor additions would substantially strengthen this paper.

Major

1.       Reference to similar, recent reviews may help the interested reader (PMID: 34117715; PMID: 23420518; DOI:10.1211/PJ.2019.20206478).

a.       I’d also recommend reviewing this literature

2.       Throughout the paper, a substantial emphasis is placed on the role of pricing in the development of a boosting strategy. In fact, this is even included in their ranking algorithm.

a.       However, I feel that this is reasonably inappropriate because the pharmaceutical manufacturer could react to the development of a boosting strategy for their drug. It's also unclear who would ultimately be the "winner" if such a strategy was adopted (e.g., hospital, insurance, patient, company).

                                                               i.      For example, efforts to decrease the dosing interval for ibrutinib were met with blatant obstruction from the manufacturer (i.e., they changed the pricing structure. (PMID: 30254130)).

3.       Similarly, emphasis is placed on the available dosage forms. However, manipulation of dosage forms (e.g., crushing tablets) could overcome issues with flat-based pricing.

a.       For example, something similar has previously been done with sorafenib (PMID: 23788485).

4.       Throughout the paper, the idea that kinase inhibitors have overlapping indications is neglected.

a.       Kinase inhibitors are ranked by their “boost-ability.” However, whether the drug is likely to remain “first-in-class” should likely be a consideration in the development of a boosting strategy.

                                                               i.      This is especially important considering a boosting trial for crizotinib was terminated due to the advent of alectinib

b.       For example, the development of a boosting strategy for ibrutinib is likely to be relatively less desirable with the advent of more effective drugs (e.g., acalabrutinib)

c.       I realize that incorporating this idea into the ranking system would be challenging, but at least recognizing this idea would be helpful to guide someone aspiring to develop this strategy for an “ideal” kinase inhibitor

5.       When discussing the decrease in interpatient variability with increasing bioavailability, you’d be remiss to neglect the classical study on this idea (PMID: 8988062)

6.       This paper focuses on boosting of drugs through CYP3A inhibition. However, the potential role of drug transporters is reasonably neglected throughout the paper.

a.       This is especially important considering the known role of p-gp in the disposition of many of these drugs and the known inhibition of p-gp (and related transporters) by these boosting agents

                                                               i.      This could also alter the distribution of the kinase inhibitor, resulting in unexpected toxicities (i.e., as described here PMID: 34117715)

b.       Nonetheless, a recent study demonstrated that cobicistat-induced increases of ibrutinib exposure were primarily mediated by CYP3A (PMID: 34950932)

                                                               i.      Without this level of mechanistic detail, it’s challenging to definitively state that the boosting of each of these kinase inhibitors is occurring exclusively through CYP3A inhibition

                                                             ii.      This paper also found that the DDI between IBR and COBI was nonlinear (as in line 640)

c.       I’d consider recognizing this idea (i.e., drug transporters may be playing a role) and perhaps adding a column to Table 3 outlining the drug transporters relevant to each TKI (perhaps at least the efflux transporters).

7.       Despite being cited, it’s unclear why this study is not described more thoroughly (PMID: 26858332)

8.       I appreciate the discussion of TDM, but I find this to be awkward (paragraph line 698). Despite vaguely outlining three scenarios, the general conclusion is that TDM would be helpful (which is certainly true). I would reorient this discussion to include a comparison of boosting versus TDM and focus on how TDM could complement a boosting strategy.

a.       I’d also include a few negatives about TDM (e.g., lacking infrastructure, logistic difficulties, cost) to support the development of boosting strategies sans TDM

b.       Without providing examples of kinase inhibitors without an exposure-response relationship (including compelling evidence of this), I find it hard to believe that many of these drugs lack this relationship. Surely this relationship must exist, at least to some degree. However, if it doesn't, then this seems critical to include in your ranking of which TKIs are likely to benefit from boosting (i.e., if there is no exposure-response relationship, then it's unlikely that boosting would be useful). This discrepancy should be navigated.

9.       When boosted, some kinase inhibitors have increased variability. I find this idea interesting and to support the idea that some kinase inhibitors are suboptimal for the development of this type of strategy.

10.   Many of these “boosting agents” have toxicities of their own (e.g., liver toxicity with ketoconazole, QT prolongation with itraconazole). However, antifungal prophylaxis could be a benefit for many cancer patients. I'd quickly discuss these advantages/disadvantages in the design of strategies to boost kinase inhibitors.

a.       Addition of a boosting agent could also increase the pill burden for patients that this is already an issue for

11.   Many of these TKI’s are autoinhibitors/autoinducers of their own transport/metabolism (e.g., imatinib inhibits CYP3A), which could complicate the development of this type of strategy for these drugs

12.   It’d be great to include bioavailability data for the drugs for which this information is available (e.g., imatinib, ibrutinib)

13. Chemical strategies to increase the bioavailability of kinase inhibitors are in development (e.g., nanoparticle formulations, pro-drugs). I think you should perhaps provide an example or two and at least state that these are outside the scope of this paper. 

14. The successful adoption of a boosting regimen for anticancer drugs that are not kinase inhibitors (e.g., decitabine/cedazuridine) reasonably supports the feasibility of developing this strategy for kinase inhibitors. Albeit, with obvious differences.

Minor

·         Line 601-613: I find this discussion to be awkward. It would seem that adding a drug that inhibits CYP3A to a CYP3A boosting regimen would be fine (i.e., CYP3A is already inhibited). I think this might be better if reoriented to discuss adding independent drugs that are CYP3A substrates to a CYP3A boosting regimen (e.g., adding atorvastatin to a cobicistat/ibrutinib regimen could be problematic).

·         Line 205: Ritonavir has been shown to inhibit OATP1B1 (PMID: 35456528).

o   Cobicistat is a reasonably well-described irreversible inhibitor of CYP3A (PMID: 21348537)

·         Line 66: Van vs Can

·         Table Line 558: Itraconazole should be capitalized

Reviewer 3 Report

The manuscript is well written and is of significance. There are a few issues needed to be addressed. 

1. Page 3, the authors defined 0, 1, 2 to reflect the contribution of metabolites. How do the authors confirm whether the metabolites were experimentally reported or from prediction?

2. Page 4: The authors stated “For publica-120 tions about CYP3A4 boosting we used the following search query in Pubmed”, why using Pubmed to check for drug-drug interactions or drug-food interaction, or CYP3A4 interactions? Why not used drugbank.ca or SuperCYP database?

3. Page 7-10, Table 3, how did the authors make sure the list of CYP enzymes are completed under the column of Metabolism for each listed drug?

Round 2

Reviewer 2 Report

This is an excellent paper that brings me joy. Edits were made consistent with suggestions and exceeded expectations.